It’s all about time: precision and accuracy of Emotiv event-marking for ERP research

http://orcid.org/0000-0003-4349-906X Williams Nikolas S. 1 nikolas.williams@mq.edu.au
http://orcid.org/0000-0003-1912-820X McArthur Genevieve M. 1
http://orcid.org/0000-0001-6862-4694 Badcock Nicholas A. 1 2
1 Department of Cognitive Science, Macquarie University , Sydney, NSW , Australia
2 School of Psychological Science, University of Western Australia , Perth, WA , Australia
Gollo Leonardo
Electronic publication date: 2021 Feb 9
Publication date: 2021
Volume: 9
Electronic Location ID: e10700
Received 2020 Oct 8; Accepted 2020 Dec 14
Copyright: © 2021 Williams et al.
Copyright year: 2021
Copyright holder: Williams et al.
License: This is an open access article distributed under the terms of the Creative Commons Attribution License, which permits unrestricted use, distribution, reproduction and adaptation in any medium and for any purpose provided that it is properly attributed. For attribution, the original author(s), title, publication source (PeerJ) and either DOI or URL of the article must be cited.
License URL: https://creativecommons.org/licenses/by/4.0/

Keywords: EEG, ERP, Emotiv, EPOC, Flex, Jitter, Timing, Accuracy, Trigger

Funding: Macquarie University and Emotiv Pty Ltd 83673928 This work was supported by the Neural Markers of Learning Success industry partnership grant (No. 83673928) between Macquarie University and Emotiv Pty Ltd. The funders had no role in study design, data collection and analysis, decision to publish, or preparation of the manuscript.

==============================
Background

The use of consumer-grade electroencephalography (EEG) systems for research purposes has become more prevalent. In event-related potential (ERP) research, it is critical that these systems have precise and accurate timing. The aim of the current study was to investigate the timing reliability of event-marking solutions used with Emotiv commercial EEG systems.

Method

We conducted three experiments. In Experiment 1 we established a jitter threshold (i.e. the point at which jitter made an event-marking method unreliable). To do this, we introduced statistical noise to the temporal position of event-marks of a pre-existing ERP dataset (recorded with a research-grade system, Neuroscan SynAmps2 at 1,000 Hz using parallel-port event-marking) and calculated the level at which the waveform peaks differed statistically from the original waveform. In Experiment 2 we established a method to identify ‘true’ events (i.e. when an event should appear in the EEG data). We did this by inserting 1,000 events into Neuroscan data using a custom-built event-marking system, the ‘Airmarker’, which marks events by triggering voltage spikes in two EEG channels. We used the lag between Airmarker events and events generated by Neuroscan as a reference for comparisons in Experiment 3. In Experiment 3 we measured the precision and accuracy of three types of Emotiv event-marking by generating 1,000 events, 1 s apart. We measured precision as the variability (standard deviation in ms) of Emotiv events and accuracy as the mean difference between Emotiv events and true events. The three triggering methods we tested were: (1) Parallel-port-generated TTL triggers; (2) Arduino-generated TTL triggers; and (3) Serial-port triggers. In Methods 1 and 2 we used an auxiliary device, Emotiv Extender, to incorporate triggers into the EEG data. We tested these event-marking methods across three configurations of Emotiv EEG systems: (1) Emotiv EPOC+ sampling at 128 Hz; (2) Emotiv EPOC+ sampling at 256 Hz; and (3) Emotiv EPOC Flex sampling at 128 Hz.

Results

In Experiment 1 we found that the smaller P1 and N1 peaks were attenuated at lower levels of jitter relative to the larger P2 peak (21 ms, 16 ms, and 45 ms for P1, N1, and P2, respectively). In Experiment 2, we found an average lag of 30.96 ms for Airmarker events relative to Neuroscan events. In Experiment 3, we found some lag in all configurations. However, all configurations exhibited precision of less than a single sample, with serial-port-marking the most precise when paired with EPOC+ sampling at 256 Hz.

Conclusion

All Emotiv event-marking methods and configurations that we tested were precise enough for ERP research as the precision of each method would provide ERP waveforms statistically equivalent to a research-standard system. Though all systems exhibited some level of inaccuracy, researchers could easily account for these during data processing.

Introduction

The use of consumer-grade electroencephalography (EEG) devices has increased markedly in recent years. EEG devices measure the voltage of electrical fields generated when neurones fire and whereas early EEG systems were cumbersome and expensive, newer systems have become smaller and cheaper. This is particularly true of commercial-grade EEG. These systems have lowered the financial barrier to neuroscientific research and, due to their portable nature, allowed studies to move outside the laboratory into more naturalistic settings, such as the classroom (see Xu & Zhong (2018) for a review). Even when used in a laboratory, commercial EEG systems can streamline data collection as the setup is often quicker and simpler than traditional EEG systems.

Research techniques that were once possible only with expensive EEG setups are now achievable using low-cost alternatives (Sawangjai et al., 2020; Williams, McArthur & Badcock, 2020). One of these techniques is the event-related potential (ERP) approach. An ERP is the average electrical potential generated by large groups of neurons in response to a particular event. It is measured by recording a person’s EEG during the repeated occurrence of a stimulus and then isolating the EEG into discrete sections of time, or epochs. These epochs contain the neural response of interest to each individual event and are averaged together to produce an ERP (see Fig. 1B, for a typical auditory ERP).

Figure 1 The effects of increasing event-marking jitter on an exemplary ERP waveform.

(A) Mean P1, N1, and P2 peak values for increasing levels of jitter (in ms SD). Open circles represent the mean peak values at each jitter level. Bars represent 95% confidence intervals. The rectangular shaded areas represent the 95% confidence interval of the original waveform peak. (B) The original ERP waveform and the effects of 10, 20, 30, 40, and 50 ms SD of jitter.

A number of studies have validated commercial-grade EEG devices for ERP research by comparing their performance to research-grade systems (see Sawangjai et al., 2020 for a review). Overall, the results have been encouraging. For example, Krigolson et al. (2017) found that a MUSE EEG system could measure ERP components in a visual oddball and a reward-learning task. Similarly, Emotiv’s EPOC system was found to measure research-grade auditory ERPs in adults (Badcock et al., 2013) and children (Badcock et al., 2015); as well as visual ERPs in response to faces (De Lissa et al., 2015). Recently, Williams et al. (2020) found analogous results for the Emotiv EPOC Flex system. The fact that EEG systems in this class cost a fraction of the price of research systems makes them an appealing alternative to researchers for ‘acquiring research-grade ERPs on a shoestring budget’ (Barham et al., 2017).

To capitalise on the ERP technique, it is critical to know exactly when a stimulus occurs. This is because the EEG signal of interest occurs very quickly following the stimulus—often under 300 ms. To accurately represent the signal requires a method of incorporating precise stimulus timestamps, or events, into EEG data in order to isolate epochs. If events are inserted at the wrong time, then the epochs do not represent the desired signal. Further, if events are inserted at varying incorrect times relative to the stimulus then the result is often severely degraded or non-existent ERPs (see Fig. 1B for an example of a degraded waveform).

Before going further, we address the use of the terms ‘trigger’ and ‘event’ in ERP research. Many studies use the two terms interchangeably. However, for clarity we draw a distinction. We use the term ‘trigger’ to denote the production of some signal (e.g. TTL pulse) that is indicative of the time a stimulus occurred and is transmitted to the EEG data. We use the term ‘event’ to denote the timestamped incorporation of that signal into the data. Thus, an experimental stimulus script (e.g. MATLAB) generates a trigger (e.g. TTL pulse), which is then received as an event in the EEG data.

An obstacle in ERP research using commercial-grade EEG devices is time-locking the stimulus with the EEG data to derive ERP components. This is because these systems were not designed for ERP research and often do not have in-built methods for event-marking. Even in cases in which there exist event-marking solutions, the results can be inconsistent. For example in early iterations of Emotiv software researchers have found that serial-port-based event-marking was unreliable and did not produce quality ERPs (Hairston, 2012; Ries et al., 2014). Researchers have attempted to circumvent this problem using various methods. Some have used offline processing techniques such as regression-based timing correction of triggers (Akimoto & Takano, 2018; Whitaker & Hairston, 2012), or using the timestamps from the log files of the stimulus scripts (Hairston, 2012; Ries et al., 2014). Others have approached this issue by using a custom-built event-marking system (the ‘Airmarker’) that converted an audio or visual stimulus into an infrared light pulse (Thie, 2013). This pulse was then transmitted to a custom-built receiver mounted on a portable EEG device (Emotiv EPOC in this case) and injected into two of the EEG channels (for a full description of procedure and equipment see Badcock et al. (2015), Thie (2013)). Events were thus visualised as distinct voltage spike in the EEG signal and timing of the events was calculated according to the onset of the spikes. While this approach yielded ERPs, it required post-processing and the sacrifice of two EEG channels. Thus, a dedicated system that incorporates events directly into EEG data would be preferable to an alternative that requires fabrication of a custom device, modification to an EEG system, and substantial post-processing.

Though previous iterations of Emotiv EEG acquisition software were unreliable for event-marking, the situation may be improved by developments in hardware and software. Hardware-based event-marking can now be achieved using a device called Extender. Likewise, serial-port event-marking is purported to be more reliable with version 2 of Emotiv Pro software relative to earlier Emotiv acquisition software such as Testbench or Emotiv Pro version 1. While these options promise to deliver synchronisation of stimulus presentation and EEG data, their reliability is untested.

For an event-marking system to reliably produce ERPs, it must be both accurate and precise. Accuracy refers to the time difference between when an event is received in the EEG data (e.g. parallel-port code received) and when the respective stimulus actually occurs (e.g. audio tone is emitted from a speaker). This is often referred to as the ‘lag’. Precision refers to the variability in the accuracy of the event-mark and is often referred to as ‘jitter’. As an example, consider a system that generates audio tones and in which the event-mark consistently appears in the EEG data 20 ms after the sound comes out of a speaker. This 20 ms difference is considered the lag and can easily be accounted for during post-processing by subtracting 20 ms from each event. However, if the difference is sometimes 12 ms, sometimes 27 ms, sometimes 33 ms, etc., this is considered imprecise, or ‘jittery’, timing. Jittery timing is difficult to correct as the difference between the stimulus and event-mark is unknown from trial to trial. A jittery event-marking system is problematic for deriving ERPs as it may distort the averaged component. For example, Hairston (2012) simulated the effect of 55 ms of timing jitter on an ERP and found that the waveform was almost entirely attenuated. Likewise, a study by Ries et al. (2014) presented results from an Emotiv device with jittery event-marking that showed severe waveform degradation compared to the waveform when the timing was corrected. Thus, ERP researchers can account for inaccurate triggers but not for imprecise ones.

Though jitter in an event-marking system is more problematic than inaccuracy, it is easier to measure. It can be quantified as the variability (e.g. standard deviation) of known inter-trial intervals (the time difference between the events). For example, if successive stimuli are presented 1,000 ms apart, then a perfect system would exhibit a mean inter-trial interval of 1,000 ms and a standard deviation of 0 ms. This would indicate that each event was recorded precisely 1,000 ms after the preceding event.

Accuracy, though less problematic than imprecision, is more difficult to measure. This is because one must know when an event should occur in the EEG data in order to compare when the event actually does occur. That is, how closely in time does the actual event match up to the EEG signal of interest. There are various methods for assessing accuracy but most include inserting some stimulus-related signal into the EEG. One example is inserting the signal from a microphone positioned by a speaker into an EEG channel. This would provide a visual reference in the EEG of when the stimulus (e.g. audio tone) occurred.

With these considerations in mind, the aim of this study was to quantify the timing of Emotiv hardware and software used for ERP research. We conducted three experiments in which we examined both the accuracy and precision of event-marking timing. In Experiment 1, we established a jitter threshold by introducing temporal noise into the events-marks of a pre-existing, exemplary ERP dataset (Badcock et al., 2013) collected with a research-grade EEG system, Neuroscan, and calculating the jitter levels at which the ERP waveform peaks were statistically different to the exemplar.

In Experiment 2, we benchmarked a method, for use in Experiment 3, to assess the accuracy of Emotiv event-marking. As previously noted, assessing the accuracy of event-marking entails measuring the time difference between when an event should occur and when it actually does occur. This is problematic as it requires something to represent the ‘true’ event. We approached this problem by measuring the lag of Airmarker events relative to a research-grade system, Neuroscan with parallel-port event-marking. We attributed this lag to Airmarker processing time and subtracted it from Airmarker events in Experiment 3 to calculate the true event times (i.e. when an event should have occurred; See Fig. 2).

Figure 2 Visualisation of how ‘true’ event times were determined in Experiment 3.

True events in Experiment 3 were calculated by subtracting Airmarker processing time observed in Experiment 2 (30.96 ms) from Airmarker events in Experiment 3. Please note that x- and y-axis values are not provided because Fig. 2 is for visualisation and clarity only. Actual timing calculations appear in Table 1.

In Experiment 3, we measured the precision of both hardware-based (i.e. Emotiv Extender) and software-based (i.e. serial port) events. We used the thresholds established in Experiment 1 to determine whether the event-marking methods were sufficiently precise. To assess accuracy, we compared Emotiv events to true event times, which were calculated as simultaneously-generated Airmarker events minus the average lag calculated in Experiment 2.

Experiment 1: establishing jitter thresholds

The purpose of Experiment 1 was to determine the tolerance of an ERP to jitter. To investigate this, we used a single pre-existing dataset selected because it exhibited a classic auditory ERP with standard P1, N1, and P2 peaks. We then incrementally introduced random noise, or jitter, into the event-marks. This allowed us to calculate jitter thresholds by establishing the tolerance of an ERP waveform to timing imprecision. Data and processing and analysis scripts may be found at Open Science Framework (https://osf.io/pj9k3/).

Materials and Methods

An EEG datafile was taken from an auditory oddball validation study (for complete details see, Badcock et al., 2013) in which participants heard 666 tones. Of these, 566 were standard (1,000 Hz) and 100 were deviant (1,200 Hz) 175 ms pure tones, with an inter-tone onset interval that randomly varied between 900 and 1,100 ms. Participants watched a silent DVD while listening to tones. EEG data were collected with Neuroscan SynAmps2 using Scan software (4.3), recorded at 1,000 Hz from 16 electrodes: F3, F7, FC4, FT7, T7, P7, O1, O2, P8, T8, FT8, FC4, F8, F4, M1 (online reference), and M2; with VEOG and HEOG; and the ground at AFz. The tone onset was marked in the EEG data via parallel port using Presentation (version 16; Neurobehavioral System Inc., Berkeley, CA, USA).

Processing and analysis

We used a single electrode for the current purposes, F3, selected for having a clear ERP waveform. We selected an individual with clearly defined P1, N1, and P2 peaks in response to standard tones. The processing was conducted as in Badcock et al. (2013) with the exception that the data were not downsampled (processing included 0.1 and 30 Hz bandpass filters, independent components analysis removal of eye-blink artefacts, epoching −100 to 600 relative to tone onset, and baseline correction). All processing was conducted with EEGLAB 14.1.0b (Delorme & Makeig, 2004). Epoching and baseline correction were repeated at different levels of temporal jitter of the parallel-port event mark. A value of 0 reflects no adjustment. We then jittered the temporal position of the events by generating a normal distribution with a standard deviation of increasing values from 1 to 50 ms in 1 ms intervals. This resulted in 50 discrete jitter levels. To remove any potential EEG artefacts, a cut-off of activation beyond ±150 μV was set for epoch exclusion. No epochs were excluded at any jitter level. Peak magnitudes were determined using an automated method that selected peak values within the following time periods: P1, 36-96; N1, 75-135; P2, 140-200 ms (eventMark_EEGLAB_processing4erps.m; https://osf.io/pj9k3/). These reflected intervals of ±30 ms either side of the peak time-point for the 0 jitter waveform to the standard tone.

To calculate jitter thresholds, we performed a series of Bayesian t-tests for each peak (i.e. P1, N1, P2). These tests compared the distributions of peak values at each jitter level to the distribution of peak values of the original waveform (i.e. zero-jitter). For each peak, we deemed the threshold to be the smallest jitter level point at which the Bayes Factor exceeded a value of 3. This indicated substantial evidence that the respective peak value was statistically different from the original (Jarosz & Wiley, 2014).

Results and Discussion

Figure 1A shows the distribution of peak means at each level of jitter. The jitter thresholds differed for each of the peaks: P1 was statistically different from the original at 21 ms of jitter, BF10 = 9.63; N1 was statistically different from the original at 16 ms of jitter, BF10 = 4.92; and P2 was statistically different from the original at 45 ms of jitter, BF10 = 20.19. Figure 1B shows the waveforms produced by increasing levels of jitter. Overall, these results suggest that larger auditory ERP peaks are more resilient to jitter, whereas smaller peaks are more easily attenuated. Further, these findings provide levels at which event-marking devices become too jittery for ERP research. We note that these values should be considered guidelines and not be interpreted as absolute precision thresholds. For the purposes of the current study, they represented values against which we could compare subsequent timing analyses.

Experiment 2: establishing the airmarker benchmark

The purpose of Experiment 2 was to establish a benchmark to which we could compare the accuracy of Emotiv event-marking systems. We did this by establishing the precision and accuracy of an event-marking system previously used in our lab (Badcock et al., 2015), the Airmarker. Data, the triggering script, and processing and analysis scripts may be found at https://osf.io/pj9k3/.

Methods

The triggering script was run on a Dell Precision T3620 computer running Windows 10 version 1607. We used a custom-written MATLAB (version R2017b) script that included the Psychtoolbox plugin (Brainard, 1997; Kleiner et al., 2007; Pelli, 1997) and a plugin to interface with the parallel-port hardware (http://apps.usd.edu/coglab/psyc770/IO32.html).

The script generated 1,000 events, each 1,000 ms apart. There were two types of events: a parallel-port code sent through a Sunix LPT PCI card and a 1,000 Hz audio tone sent through a 3.5 mm audio output port. The parallel port trigger went to the Neuroscan amplifier where it was incorporated as an event into the EEG data. The audio tone fed into the Airmarker transmitter and was converted to an infrared signal that was received by the Airmarker receiver and converted to a square electrical wave. We attached the positive and negative Airmarker receiver wires to a bipolar electrode of the Neuroscan system (VEOG). We used a Neuroscan Synamps2 system at a 1,000 Hz sampling rate to collect EEG data to Curry acquisition software (version 7; compumedicsneuroscan.com) on a Dell Optiplex 7760 computer running Windows 10 version 1809. See Fig. 3 for a schematic of the triggering setup.

Figure 3 Experiment 2 event-marking setup schematic.

Processing and analysis

Electroencephalography data were imported using EEGLAB (Delorme & Makeig, 2004). To derive Airmarker triggers we wrote a custom MATLAB script that calculated the absolute value of the EEG channel derivative and then set a threshold of +3 standard deviations above the derivative mean. Within the time-window of 200 ms following each parallel-port trigger, the script identified the first sample in which the Airmarker EEG derivative exceeded the threshold. The time point of each of these samples was considered an Airmarker event. We then calculated inter-trial intervals for each event type (parallel-port and Airmarker, independently) as the time between adjacent events. Precision, or jitter, was thus quantified as the variability (i.e. standard deviation) of the inter-trial intervals within each event type. We quantified lag as the average difference between Neuroscan parallel-port events and Airmarker events. See Fig. 4 for an example of a three-trial sequence of Airmarker EEG signal and derived events with annotations depicting inter-trial intervals and lag.

Figure 4 Three-trial example of Airmarker EEG signal with parallel port and derived Airmarker events in Experiment 2.

Note that the parallel port and Airmarker events do not represent any real values on the y-axis but are presented for visualisation only. The annotations depict the relationships between events for calculating inter-trial intervals and lag.

Results

Table 1 shows the timing performance in Experiment 2 (and 3). We observed sub-millisecond precision with respect to the parallel-port trigger (Fig. 5A). The Airmarker trigger was slightly less precise (Inter-trial interval SD = 3.49 ms), though was well below the thresholds established in Experiment 1. On average, Airmarker triggers appeared in the EEG data 30.96 ms behind parallel-port triggers. As we assumed a near-zero latency for parallel-port triggers in the Neuroscan configuration, we considered 30.96 ms the processing lag associated with the Airmarker and subtracted this calculation from Airmarker lag times in each configuration in Experiment 3. This allowed us to examine the accuracy of Emotiv event-marking.

Table 1 Precision and accuracy of event-marking methods in Experiments 2 and 3.

Jitter was calculated as the standard deviation (in samples and ms units) of the inter-trial intervals. Lag was calculated as the average time difference (in ms) between the tested event-marking systems and ‘true’ events. True events were calculated as Airmarker times minus Airmarker processing time (30.96 ms). Negative lag values indicate events arrived earlier than true events in the data, whereas positive lag values indicate events arrived later than true events in the data.

EEG System	Sampling rate (Hz)	Trigger Method	Dropped samples	Missed triggers	Jitter (samples SD)	Jitter (ms SD)	Lag (ms)	
Neuroscan	1,000	Parallel port	–	–	0.43	0.43	–	
		Airmarker	–	–	3.49	3.49	30.96	
EPOC+	128	Parallel port to Extender	56	0	0.53	4.14	−57.1⊥	
		Arduino to Extender	112	1	0.64	4.97	−52.37⊥	
		Serial port	42	0	0.54	4.21	−22.29⊥	
	256	Parallel port to Extender	0	0	0.72	2.83*	−55.45⊥	
		Arduino to Extender	0	0	0.98	3.83*	−51.61⊥	
		Serial port	0	0	0.49	1.91*	10.54⊥	
Flex	128	Parallel port to Extender	21	1	0.43	3.39	−56.01⊥	
		Arduino to Extender	21	0	0.58	4.57*	−52.09⊥	
		Serial port	7	0	0.45	3.48	−19.66⊥	
Notes:

* The jitter of a configuration was statistically different (p < 0.05) from other configurations within that device.

⊥ Lag measures have been corrected by subtracting Airmarker processing time (30.96 ms).

SD, standard deviation.

Figure 5 Boxplots of the inter-trial intervals observed for each triggering method in Experiments 2 and 3.

(A) Parallel-port triggering with Neuroscan SynAmps2 acquired with Curry Software. (B) Arduino-generated TTL triggers to Emotiv Extender acquired with Emotiv Pro. (C) Parallel-port-generated triggers to Emotiv Extender acquired with Emotiv Pro. (D) Serial-port-generated triggers acquired with Emotiv Pro.

Experiment 3: emotiv and airmarker triggering

The purpose of Experiment 3 was to examine the accuracy and precision of ERP triggers with Emotiv EEG hardware. To do this, we tested three event-marking methods: (1) a parallel-port-generated TTL trigger sent to Emotiv Extender hardware (Extender); (2) an Arduino-generated TTL trigger sent to Extender; and (3) a serial-port-code trigger sent directly to the acquisition computer. For each of these methods we tested three Emotiv EEG configurations: (1) Emotiv EPOC+ (EPOC) at 128 Hz sampling rate; (2) EPOC at 256 Hz sampling rate; and (3) Emotiv EPOC Flex (Flex) at 128 Hz sampling rate. Data, the triggering script, and processing and analysis scripts may be found at https://osf.io/pj9k3/.

Methods

The stimulus and acquisition computers were the same as in Experiment 2. We also used the same triggering script as Experiment 2 in which audio-tone triggers were sent to Airmarker. To incorporate Airmarker events into the Emotiv EEG data, we used the same procedure as in previous validation studies (Badcock et al., 2013, 2015; De Lissa et al., 2015). We connected the receiver wires to two channels of the Emotiv device and biased them to the driven-right-leg (DRL) channel using a second set of wires that included a 4.7 kΩ resistor (Fig. 6D). This setup is necessary with ‘active’ EEG systems to simulate a connected head circuit and obtain a clean EEG signal. See Fig. 6 for a schematic of the parallel-port trigger (A), Arduino Uno trigger (B), and serial-port trigger (C), configurations.

Figure 6 Experiment 3 triggering setup schematics.

(A) Parallel-port generated TTL pulse to Extender. (B) Arduino-generated TTL pulse to Extender. (C) Serial-port triggering. (D) Airmarker and bias wire configuration used to insert Airmarker signal into Emotiv EEG channels.

For the TTL triggering, we wrote a switch into the MATLAB code that depended on triggering method (i.e. parallel port, Arduino, or serial port). The exact setup varied for each configuration, and each is described below. In each case, we generated 1,000 triggers, 1,000 ms apart. All EEG data were acquired using Emotiv Pro (2.3.0). The triggering script may be found at https://osf.io/pj9k3/.

Parallel port to extender

To generate parallel-port TTL triggers, we used the same plugin as in Experiment 2. TTL triggers were transmitted using a custom-built parallel-port-to-BNC adapter that carried the pulse from a single parallel-port pin to a 2.5 mm tip-ring-sleeve jack plugged into Extender. The event was then incorporated into the Emotiv device data (i.e. EPOC or Flex) by a USB cable where it was transmitted via Bluetooth to the acquisition computer.

Arduino to extender

For the Arduino to Extender testing we used the MATLAB Support Pack for Arduino Hardware (https://au.mathworks.com/matlabcentral/fileexchange/47522-matlab-support-package-for-arduino-hardware) that interfaced with an Arduino Uno (https://arduino.cc). Triggers were achieved by sending a digital pin output command to the Arduino, which then sent a TTL pulse to a 2.5 mm tip-ring-sleeve jack plugged into Extender. As before, the event was then incorporated into the Emotiv EEG data and transmitted to the acquisition computer via Bluetooth.

Serial port

To generate serial-port-code triggers, we used native MATLAB functions. The trigger was sent from a serial port to a virtual serial-port USB adapter on the acquisition computer. Serial-port events were then incorporated directly into the Emotiv EEG data in Emotiv Pro.

Processing and analysis

Electroencephalography data were imported using EEGLAB (Delorme & Makeig, 2004). We first calculated the number of dropped samples in each configuration. We did this because wireless EEG systems, like EPOC and Flex, can sometimes experience interference that results in incomplete data transmission. To calculate the number of dropped samples, we counted the number of instances in which a value of ‘1’ appeared in the ‘INTERPOLATION’ channel. This indicates that the acquisition software did not receive a sample and thus interpolated EEG channel values according to temporally-adjacent channel values. We also calculated the number of times dropped samples resulted in missed triggers. Though it was rare, this situation did arise in two configurations.

We again calculated the inter-trial intervals for each of the primary triggering methods and used the standard deviation as a measure of precision. In the configurations where a trigger was missed, we removed the affected inter-trial intervals before calculating timing numbers.

We calculated Airmarker events identically to Experiment 2, using the +3 standard deviation above the mean method. For each configuration we also calculated a measure of accuracy by first determining the true event time. The true event time was calculated by subtracting 30.96 ms (i.e. the Airmarker processing time measured observed in Experiment 2) from Airmarker events (see Fig. 2). Emotiv event-marking accuracy was then calculated as the average time difference between Emotiv events and the true events.

Results

Table 1 shows the timing results. Overall, Emotiv triggering systems were well below Experiment 1 thresholds. To compare jitter between triggering systems within each device configuration, we performed Levene’s tests of equality of variance on the inter-trial intervals with follow-up pairwise comparisons (Bonferroni corrected for the number of comparisons) where we detected significant results. Results of the EPOC 128 Hz configuration indicated significant differences in variances (F = 3.15, p = 0.043). However, none of the follow-up tests achieved significance at the corrected α = 0.016 level (all Fs < 5.56, all p > 0.018). Results of the EPOC 256 Hz configuration indicated a significant difference in variances (F = 134.48, p < 0.001). All follow-up tests were significant at the corrected level (all Fs > 27.75, all p < 0.001) suggesting that the serial-port event-marking was the most precise, followed by parallel-port event-marking, and then Arduino-triggered event-marking. The results of the Flex configuration were also significant (F = 17.78, p < 0.001) with follow-up tests suggesting that Arduino triggering was more jittery than both parallel-port (F = 27.79, p < 0.001) and serial-port (F = 21.89, p < 0.001) event-marking. There was no difference between parallel-port and serial-port event-marking (F = 0.42, p = 0.518). Overall, these results suggested that serial-port event-marking with EPOC at 256 Hz sampling rate was the most precise configuration. We note, however, that all configurations exhibited jitter of less than a single sample. See Figs. 5B and 5C for distributions of inter-trial intervals for each configuration.

All configurations exhibited some level of inaccuracy (see Table 1). This lag indicated the difference between the event timestamp (i.e. when the stimulus was said to have occurred) and the EEG data of interest (i.e. when the Airmarker signal appeared in the EEG). We provide the calculations here for reference but note that we did not perform statistical tests on the lag measure for two reasons. The first is that the variance of this calculation is directly impacted by the precision of the event-marking trigger, which we assessed above. The second is that we do not want to give the impression that this measure would be identical in the setups of prospective users. We stress that researchers should test the accuracy of their respective configurations.

Discussion

In this study we examined the timing performance of event-marking solutions used with Emotiv EEG systems. We first established jitter thresholds by introducing noise into an exemplary ERP dataset and determining at which level the waveform was attenuated to the extent that it no longer resembled the original. We then benchmarked a custom-built event-marking system known to produce valid ERPs (i.e. the Airmarker; Thie, 2013). Finally, we used this system to identify when events should appear in the data in order to assess the timing performance of the Emotiv triggering systems.

Our first main finding was that large peaks in our ERPs were more resilient to jitter attenuation than were the smaller peaks. In Experiment 1, it took twice as much jitter to attenuate the large P2 peak as it did to attenuate the smaller P1 and N1 peaks. It is notable that jitter did not change the timing of the peaks. Rather, it suppressed the amplitude of the peaks and distorted the slopes. This is a similar pattern to previous work in which Hairston (2012) reported the effects of timing jitter on a simulated ERP. Naturally, we would not expect the thresholds we calculated in Experiment 1 to identically correspond to data collected from other individuals. Inter-individual variation in EEG signatures would certainly create idiosyncratic jitter levels of waveform attenuation. For example, data from individuals with smaller peaks would be more susceptible to waveform degradation. Likewise, other experimental factors such as high-frequency noise would influence jitter-induced attenuation.

Our second main finding was the Emotiv event-marking systems we tested were precise, with all configurations showing less than a sample of jitter. We deemed these systems sufficiently precise for ERP research based on the thresholds we established in Experiment 1. We note, however, that these thresholds were based on the ERP data from a task that produces a very robust waveform. The precision thresholds would likely vary depending on the ERP being examined. Short-latency ERPs, such as auditory brainstem responses, would be particularly susceptible to jitter-induced attenuation as they typically occur in the first 10 ms following stimulus onset with peaks lasting 1–2 ms each (Luck, 2014). However, given that the maximum sampling rate of the equipment we tested was 256 Hz, the capacity to measure auditory brainstem responses would not be limited by event-marking precision but rather by the EEG hardware. For most long-latency responses (i.e. those over 50 ms), the configurations we tested in this study would suffice.

We also found that the accuracy of event-marking systems varied, though configurations involving Extender all showed lag between 51 and 58 ms. This is in line with Extender performance expectations as the manufacturer indicates a ‘…fixed 60ms delay between the trigger and the channel signal data which is due to the filter delay for the channels.’ (https://emotiv.gitbook.io/extender-manual/hardware_triggering). Although inaccuracy is not ideal, the low levels of jitter observed across the configurations would make timing correction straightforward for ERP researchers. In line with this we note that while we provide precision and accuracy values in Table 1, we do so for reference only. We tested these systems on only one computer setup. As computer hardware and software could feasibly influence performance, we suggest that researchers employing these event-marking systems benchmark their respective setups.

Conclusion

All Emotiv event-marking configurations we tested were suitably precise for research involving long-latency ERPs. Though all configurations were somewhat inaccurate, these inaccuracies can easily be accounted for during data processing. We note that although we provide precision and accuracy calculations for these specific Emotiv event-marking solutions, we suggest researchers measure the precision and accuracy of their respective setups.

Beyond timing precision and accuracy, there are other considerations for researchers choosing an event-marking solution. These include financial limitations, hardware and software configurations, and ease-of-use. For example, dedicated trigger boxes used with high-end research-grade EEG setups will likely be extraordinarily precise. However, they may be prohibitively expensive for some. Likewise, Emotiv EEG event-marking can be achieved using custom-built equipment (e.g. Airmarker) or by interfacing directly with the software through a custom application programming interface. Either of these options requires significant expertise that some researchers may not have available. Given that serial-port event-making was among the most precision solutions in the current study, we would recommend it for researchers using Emotiv EEG as it best balances convenience, usability, and precision.

Additional Information and Declarations

Competing Interests

Author Contributions

Data Availability

Nikolas S. Williams is employed on a research fellowship that is funded by an industry partnership grant between Macquarie University and Emotiv. Genevieve McArthur and Nicholas Badcock are Academic Editors for PeerJ.

Nikolas S. Williams conceived and designed the experiments, performed the experiments, analysed the data, prepared figures and/or tables, authored or reviewed drafts of the paper, and approved the final draft.

Genevieve M. McArthur conceived and designed the experiments, authored or reviewed drafts of the paper, and approved the final draft.

Nicholas A. Badcock conceived and designed the experiments, authored or reviewed drafts of the paper, and approved the final draft.

The following information was supplied regarding data availability:

The raw data, triggering script, and processing and analyses scripts is available at the Open Science Framework:

Williams, Nikolas S, and Nicholas A Badcock. 2020. ‘Precision and Accuracy of Emotiv Event-Marking for ERP Research.’ OSF. November 2. osf.io/pj9k3.

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
