# Peer review of "It’s all about time: precision and accuracy of Emotiv event-marking for ERP research"

_PeerJ, doi:10.7717/peerj.10700_

## Round 0.1 · original submission · Minor Revisions

Your manuscript has now been seen by 2 reviewers. You will see from their comments below that they find your work of interest, and some constructive points are worth considering. We therefore invite you to revise and resubmit your manuscript, taking into account the points raised. Please highlight all changes in the manuscript text file.

·

Basic reporting

The manuscript presents an clear and careful investigation into the precision of several methods for sending event triggers to EEG hardware. The quality of the reporting is excellent.

The one aspect of the reporting that I wasn't very clear on was regarding the lag measures in Experiments 2 and 3, specifically to understand what was the ground-truth signal against which the triggers were compared to get the measure of lag. It is mentioned (Line 247 in methods of Exp 2) that an audio tone was generated for use by the Airmarker. Was the same tone replicated to one of the channels of the EEG system? The schematics don't show an audio input directly to the amps but I couldn't think how else the lag was determined. Table 1 mentions that lag was the difference between Emotiv and Airmarker events but, again, I wasn't exactly clear which carried the true timing signal. Is the "Emotiv event" that the authors mention mentioned the raw audio waveform from psychtoolbox?

Minor points:

Lines 86-90: The authors focus on the need for accuracy but I would have thought precision is the more important here (and is the topic of much of the paper). ERPs are the one place where you can sometimes see peaks under 5ms wide and such narrow peaks require very high precision to be correctly identified. Accuracy errors in the ERP can probably even be corrected post-hoc by looking at the ERP waveform and time-shifting it.

Line 103: Minor grammar error. "in which there exists" should be "in which there exist"

Line 209, 210: "A cut-off of activation beyond ±150 μV was set for epoch exclusion," is repeated

Experimental design

I believe the experimental design is perfectly sound and addressed the question appropriately.

I wondered whether the threshold for what was considered an acceptable jitter is appropriate. The authors used the point "at which jitter level the regression-line 95% confidence interval diverged from the zero-jitter 95% confidence interval." This seems an extreme statistical threshold. For comparing a variable value against a fixed value (a one-sample test) it's normal to compare against 95% CI but, in this instance, both values are variable and requiring the 95% CI on both is setting a very high threshold for what one considers a significant change, rather higher than a standard t-test for instance. In fact, I suspect a Bayesian test would be more appropriate since it doesn't seem appropriate this instance to stack the test in favour of the Null Hypoth as is being done here. Both aspects will likely reduce the level of jitter required to conclude that the values differ, although I'm sure it won't alter the *pattern* of the results described but it might make a difference to whether later measures fall within this binary threshold.

Validity of the findings

The findings seem appropriate and are reported clearly and without exaggeration.

The lags that have been measured are quite substantial, and seemingly come from a mixture of the Airmaker and the Emotiv system, but the precision is still reasonably high (given a 256 Hz sample rate) and that is the more important value of the two.

·

Basic reporting

This article is well written and clear. For example, the explanations of core concepts are explicit and concise (e.g. definition and differentiation of ‘trigger’ and ‘event as well as ‘precision’ and ‘accuracy’.). I have only 2 main comments and remaining very minor suggestions for where reporting (including figures) could be clearer.

Main comments:
- Line 263-266: This was the only area I had I re-read the definition a few times. I interpreted inter-trial-interval as the intervals within trigger type (i.e. interval from one Airmarker trigger to the next Airmarker trigger OR from one parallel-port trigger to the next parallel-port trigger). But line 263 confused me slightly, I think becasue the second sentence sounds as though it refers to the previous one, when I _think_ they are different things “We then calculated the time between each of the parallel-port and Airmarker events. The variability of these inter-trial intervals (i.e., standard deviation) represented our measure of precision (or jitter). “ I think it would help if the “inter-trial-interval/precision” and "Accuracy" was marked explicitly in Figure 3.
- Because of my misunderstanding above, I was slightly confused why the Airmarker data is not shown in Figure 4 or Table 1. For example, for Experiment 2, only the parallel port trigger data is shown in Figure 4 and Table 1, whilst the Airmarker is described in the text (line 270). It is possible that clarifying point 1 will resolve this confusion, but my initial view is that including the Airmarker data in these figs/tables would be interesting/helpful.

Minor comments:
- Line 166: perhaps provide a citation for the ‘pre-existing, exemplary’ dataset (Badcock et al., 2013) on initial mention.
- Line 190 – should there be a hyphen in 175-ms here or a space?
- Line 208 – how many sampling points were used from within the normal distribution? (i.e. how many levels of jitter)
- Line 209/210 – “A cut-off of activation beyond ±150 μV was set for epoch exclusion” is repeated twice – one can be removed.
- Why were no epochs excluded for the highest jitter level? I am assuming due to noise but maybe be explicit.
- Line 212 – it would be useful to refer to the exact supplemental file on OSF used for this automated peak detection. (as a side note, I commend the authors on their use of OSF to share their materials so clearly, a minor suggestion could be to add a README doc to the OSF page to allow readers to clearly navigate the materials)
- Figure 1 – In general this figure is very informative and thorough, but a couple of small points might help clarity.
o It would be useful to make clearer that the jitter legend corresponds to subpanel b rather than a (maybe by moving to upper right instead of top centre?).
o The shaded area around the regression line in panel a is very subtle and tricky to spot without zooming in – could the opacity of the shaded area be increased or coloured?
- Table 1 – please state in legend what the asterisks correspond to and what a negative lag equates to (it might also be standard to include definition of SD acronym?)
- Figure 5: This is a very very minor point, but it would be useful to have this larger – so a 2x2 rather that 1x4 layout could be helpful.
- Line 355 “Results of the Flex” – missing ‘the’ at start of sentence

Experimental design

The experimental design and research questions are clear and fall within the aims and scope of this journal. I have no further comment.

Validity of the findings

The data are provided and the statistical analyses are well justified. I have two main comments, one with regards to explanation of statistical results and the second regarding the discussion:

Line 350 “This indicated that there was no difference in jitter 351 between Arduino, parallel-port, and serial-port triggering” I am not sure we can go beyond stating the difference ‘failed to reach significance’ without some support for the null (e.g. a Bayesian analysis), so for now I would recommend stating the difference ‘did not reach significance’ rather than ‘no difference’.

There are some aspects I would have liked to see expanded upon a bit more in the discussion. For example:
- Experiment 1 uses a dataset from a single individual. Could the authors comment on whether they would expect variability in jitter thresholds across individuals? For example individuals with smaller, or noisier, original waveforms. On line 381 it is stated “large peaks in an ERPs are more resilient to jitter attenuation than are the smaller peaks.” Could this mean that individuals with smaller peaks (or paradigms producing less reliable or smaller peaks) would present more of an issue with jitter? If so I think this could be outlined in the discussion as it will be an important consideration for future researchers.
- Because of the point above, there are some places where I wonder if this comment ‘All Emotiv event-marking configurations we tested were suitably precise for ERP research’ needs an extension to mention that the degree of jitter may depend on the ERP being investigated.
- Importantly, the authors recommend that researchers test their respective setups to be sure of jitter with their own combination of hardware etc. Could the provided scripts be used for this? If so I think that this could be explicitly stated on line 393, as this is a further useful contribution of this work.
- On a pragmatic note, if all systems are suitable for ERP research from a jitter perspective, could the authors briefly comment what else we (researchers) could consider when picking a method? For example, because the Airmarker is a custom built system, (which presumably is not commercially available?) engineering expertise might still be required to set up the system for ERPs in other labs? But, is it possible this system is easier to engineer than other systems? This is more a point of curiosity.

Additional comments

In this article, the authors assess the precision and accuracy of ERP event marking systems for commercially available EEG (Emotiv). In Experiment 1, they assess the sensitivity of ERPs to jitter in an existing Emotiv dataset, and derive the 'jitter threshold' for a set of ERPs (how much jitter is needed before the ERP waveform is significantly degraded). In Experiment 2 they compare the 'Airmarker' event marking system with a parallel port triggers in a research grade EEG set-up in order to assess the presence of 'lag' (accuracy) and compare the variability of inter-trial-intervals (precision). In experiment 3, the inter-trial-interval of 3 event marking systems are tested with Emotiv to assess precision in each system and 'accuracy' relative to the Airmarker system. In general the jitter of the event marking systems all fell below the jitter thresholds observed in Experiment 1. Therefore the authors conclude that these event marking systems were precise enough for ERP research.

I enjoyed reading this manuscript and I believe it will be a highly useful article for those interested in using Emotiv in their research in the future. The experimental approaches appear valid and clear. My comments therefore mainly relate to minor points on phrasing and clarification.

---

## Round 0.2 · accepted · Accept

Thank you for the detailed response letter. We are delighted to accept your manuscript for publication.

·

Basic reporting

The quality of reporting was already very high and the authors have improved clarity on the key parts that had been less clear in the first iteration.

Experimental design

This was very strong in both iterations of the manuscript.

Validity of the findings

This was very strong in both iterations of the manuscript.

Additional comments

The paper describes a very useful set of measurements, and describes them very clearly.

·

Basic reporting

The authors have addressed my previous queries regarding basic reporting. The manuscript reads clearly.

Experimental design

I previously raised no further comment on the experimental design of this study as I found this clear and well defined.

Validity of the findings

I appreciate that the authors have adjusted their phrasing of effects that failed to reach significance, and that further comment has been added to whether this would generalise to other datasets, from more than one individual.

Additional comments

I thank the authors for their revisions, they have fully addressed my concerns and I recommend publication.